# Population Pharmacokinetic Analysis and Dosing Optimization of Prophylactic Fluconazole in Japanese Patients with Hematological Malignancy

**DOI:** 10.3390/jof7110975

**Published:** 2021-11-16

**Authors:** Yasutaka Sakamoto, Hikaru Isono, Yuki Enoki, Kazuaki Taguchi, Takuya Miyazaki, Hiroyoshi Kunimoto, Hirofumi Koike, Maki Hagihara, Kenji Matsumoto, Hideaki Nakajima, Yukiko Sahashi, Kazuaki Matsumoto

**Affiliations:** 1Division of Pharmacodynamics, Faculty of Pharmacy, Keio University, Tokyo 105-8512, Japan; yasutaka@yokohama-cu.ac.jp (Y.S.); hikaru.isono@z8.keio.jp (H.I.); taguchi-kz@pha.keio.ac.jp (K.T.); matsumoto-kz@pha.keio.ac.jp (K.M.); 2Department of Pharmacy, Yokohama City University Hospital, Yokohama 236-0004, Japan; hi00_k@yokohama-cu.ac.jp (H.K.); ysahashi@yokohama-cu.ac.jp (Y.S.); 3Department of Hematology and Clinical Immunology, Yokohama City University School of Medicine, Yokohama 236-0004, Japan; takuya_m@yokohama-cu.ac.jp (T.M.); kunimoto@yokohama-cu.ac.jp (H.K.); makigon@yokohama-cu.ac.jp (M.H.); makenjy@yokohama-cu.ac.jp (K.M.); hnakajim@yokohama-cu.ac.jp (H.N.)

**Keywords:** fluconazole, prophylaxis, population pharmacokinetic analysis, hematological malignancy, Monte Carlo simulation, probability of target attainment, dosing optimization, pharmacokinetics/pharmacodynamics

## Abstract

We conducted population pharmacokinetic (PPK) analysis and Monte Carlo simulations to determine the appropriate prophylactic dose of fluconazole to prevent invasive candidiasis in patients with hematological malignancies. Patients receiving chemotherapy or hematopoietic stem cell transplantation at Yokohama City University Hospital between November 2018 and March 2020 were included. Additionally, patients receiving oral fluconazole for prophylaxis were recruited. We set the free area under the curve/minimum inhibitory concentration (MIC) = 50 as the target and determined the largest MIC (breakpoint MIC) that could achieve more than 90% probability of target attainment. The blood fluconazole concentration of 54 patients (119 points) was used for PPK analysis. The optimal model was the one-compartment model with first-order administration and first-order elimination incorporating creatinine clearance (CLcr) as a covariate of clearance and body weight as a covariate of distribution volume. We conducted Monte Carlo simulation with fluconazole at 200 mg/day or 400 mg/day dosing schedules and patient body weight and CLcr ranging from 40 to 70 kg and 40–140 mL/min, respectively. The breakpoint MICs on the first dosing day and at steady state were 0.5–1.0 μg/mL and 1.0–2.0 μg/mL for 200 mg/day and 1.0–2.0 μg/mL and 2.0–4.0 μg/mL for 400 mg/day, respectively. The recommended dose was 400–700 mg/day for the loading dose and 200–400 mg/day for the maintenance dose. Our findings suggest that the optimal prophylactic dose of fluconazole in hematological malignancy patients depends on CLcr and body weight, and a sufficient loading and maintenance dose may be needed to completely prevent invasive candidiasis.

## 1. Introduction

Invasive candidiasis is one of the most common fungal infections and includes candidemia and deep-seated candidiasis [1]. Hematological malignancy is a risk factor for invasive candidiasis, and prophylaxis of this is a critical priority [2,3,4]. In a report of autopsy cases in Japan, deep-seated candidiasis was found to be the direct cause of death in 66.7% of patients with leukemia or myelodysplastic syndrome and concomitant deep-seated candidiasis [5]. According to a report on post-hematopoietic stem cell transplant recipients in the United States, one-year survival rate was only 33.6% in these patients [6]. Therefore, prevention of invasive candidiasis is a key challenge.

In the Japanese Domestic Guidelines for Management of Deep-seated Mycosis 2014, fluconazole, itraconazole, and micafungin are recommended for the prophylaxis of invasive *Candida* infection in the pre-engraftment of allogeneic transplantation, autologous transplantation, and chemotherapy-induced neutropenia [7]. Gastrointestinal absorption of itraconazole was significantly improved by modification in formulation; however, this formulation cannot be used routinely due to gastrointestinal toxicity [8]. Although micafungin is highly efficacious [9], it is not suitable for long-term prophylaxis because only the injection form is available [7]. Therefore, fluconazole is often used to prevent invasive candidiasis because of its effectiveness.

Therapeutic efficacy has been reported to be correlated with fluconazole dose [10], and hence, 400 mg/day fluconazole is recommended for patients with hematological malignancies [3,11,12]. Although, 100–200 mg/day is often administered in Japan [13,14], the adequacy of 100–200 mg/day, i.e., less than the recommended dose, has not yet been validated [15].

Population pharmacokinetic (PPK) analysis can be used to obtain the pharmacokinetic parameters required for drug dosage optimization. Several PPK analyses of fluconazole have been reported [16,17,18,19,20]. However, the PPK parameters of fluconazole in patients with hematological malignancies have not yet been clarified. Because augmented renal clearance (ARC) increases drug clearance in hematological malignancy patients [21,22], the pharmacokinetics of hematological malignancies may differ from those with other diseases. Therefore, PPK analysis using blood fluconazole concentrations from hematological malignancy patients is necessary to clarify the appropriate dosage in patients with hematological malignancies.

In this study, we built a PPK model of fluconazole in patients with hematological malignancies and investigated the optimal dosage using Monte Carlo simulation.

## 2. Materials and Methods

### 2.1. Ethics

This study was approved by the Yokohama City University Ethics Committee (approval number: B180906004) and the Keio University Faculty of Pharmacy (approval number: 190704-1). Written informed consent was obtained from all participants who were administered fluconazole at the Yokohama City University Hospital.

### 2.2. Patients

Among the patients admitted to the Yokohama City University Hospital in Japan between November 2018 and March 2020 who received chemotherapy or hematopoietic stem cell transplantation for hematological malignancy, only those who met the following criteria were enrolled in this study: age ≥ 16 years and prophylactic fluconazole administration. Patients aged < 16 years who could not provide consent or critically ill patients were excluded.

### 2.3. Fluconazole Administration and Sample Collection

Fluconazole was administered orally at a dose of 200 mg once daily every 24 h. Blood samples were collected at 1−4 points at indicated time (1 h before (trough levels), 2, 4, or 12 h) after fluconazole administration. Plasma was immediately extracted and stored at −20 °C at the hospital. Plasma was then transported to the Keio University Faculty of Pharmacy and stored at −80 °C until measurement. Clinical and patient background data on the first day of fluconazole administration were collected. Body surface area (BSA), creatinine clearance (CLcr), and estimated GFRcre (eGFRcre) were calculated using the Dubois formula [23], Cockcroft-Gault formula [24], and the Japanese-recommended eGFR equation by Matsuo et al. [25].

### 2.4. Measurement of Fluconazole Concentration

Fluconazole Standard solutions (1.56, 3.12, 6.25, 12.5, 25.0, 50.0 μg/mL of fluconazole standards (FUJIFILM Wako Pure Chemical Corp. Osaka, Japan)) or samples were mixed with dichloromethane, and the organic phase was evaporated and then redissolved in the mobile phase where the resulting solution was used as a sample. A calibration curve was prepared using the standard solution, and the concentration of the sample was measured. Fluconazole concentrations were measured using high-performance liquid chromatography (HPLC; Shimadzu, Kyoto, Japan). The HPLC conditions were as follows: Mightysil^®^ RP-18 GP 250–4.6 (5 μm) analytical column was used, the measurement temperature was 40 °C, the mobile phase was 10 mM phosphate buffer (pH 5.70): methanol = 65:35, and the wavelength of the ultraviolet absorption detector (SPD-20A) was set to 266 nm.

### 2.5. PPK Analysis

PPK analysis was performed using the Phoenix NLME™ (Certara, Princeton, NJ, USA). First-order conditional estimation-extended least squares (FOCE-ELS) methods that are equivalent to the NONMEM FOCE methodology with interaction (FOCE-I) were used throughout the modeling process. First, one-, two-, and three-compartment models with first-order administration and first-order elimination were tested to determine fluconazole distribution. The PK models were assessed using statistical methods. Additive, multiplicative, additive and multiplicative models were tested to describe the residual variability model. The optimal base compartment model was evaluated for statistical significance using the objective function value (OFV; −2 Log-likelihood, −2LL) and Akaike information criterion. Subsequently, the final model was evaluated, including covariates that could affect fluconazole PK parameters. The evaluated covariates were age, body weight, height, BSA, body mass index, CLcr estimated using the Cockcroft-Gault formula, and eGFRcre. The covariates were tested using a stepwise forward inclusion and backward elimination model building process.

The suitability of the final model was verified using the visual predictive check (VPC) and bootstrap methods (1000 bootstrap runs).

### 2.6. Probability of Target Attainment (PTA) Analysis

In the PTA analysis, we randomly generated data for 10,000 simulated patients by CLcr and body weight using Monte Carlo simulations and calculated the area under the curve (AUC). Phoenix ^®^ NLME™ (Certara, Princeton, NJ, USA) and Phoenix WinNonlin™ (Certara, Princeton, NJ, USA) were used. Following previous reports, the *f* AUC/minimum inhibitory concentration (MIC) and the protein binding rate were set at 50 [16,26] and 12%, respectively. The largest MIC that achieved a PTA ≥ 90% was defined as the breakpoint MIC.

Twenty-four patient backgrounds were assumed: four weights (40, 50, 60, and 70 kg) and six CLcr (40, 60, 80, 100, 120, and 140 mL/min). Fluconazole (200 mg and 400 mg, once daily) was simulated.

### 2.7. Verification of the Optimal Dose of Fluconazole

To determine the prophylactic dose, the target MIC was set at 2.0 μg/mL with reference to the sensitive norm of *Candida albicans* in CLSI M60 1st Edition [27] and previous epidemiological reports [28,29]. The loading and maintenance doses were determined for every 100 mg change. The dose in relation to body weight and CLcr, which ensured PTA ≥ 90% for *f* AUC/MIC = 50, was determined by performing Monte Carlo simulations.

## 3. Results

### 3.1. Patient Characteristics

The characteristics of the 54 patients are listed in Table 1. The median age was 53 (20–77) years, and 31 of among these were male. Median body weight, BSA, CLcr, and eGFRcre were 57.6 (39.8–99.1) kg, 1.62 (1.31–2.23) m^2^, 87.1 (31.1–193.6) mL/min, and 71.2 (25.9–148.5) mL/min, respectively. Patients with many types of hematological malignancies were included, and post-transplantation cases were included in this analysis. The fluconazole dose was 200 mg/day in all patients.

### 3.2. Validation of Fluconazole Measurement

The correlation coefficient of the calibration curve was 1.0. The accuracy of the assay ranged from 86.9% to 100.7%. The %CVs of the intraday and inter-day assays using the four test samples were <5.3% and <7.5%, respectively.

### 3.3. Pharmacokinetic Model Buiding

A total of 125 blood samples were collected from patients, and 119 were used for analysis, excluding six points for which renal function changed substantially during administration. The results of the final model and bootstrap method performed using PPK model of fluconazole are shown in Table 2. The model goodness of fit was improved by inclusion of the covariates measured CLcr (normalized to the population median of 5.2 L/h) for fluconazole clearance and body weight (normalized to 57.6 kg) for fluconazole V/F. Furthermore, the addition of both resulted in a statistically significant improvement in the Log-likelihood from the previous model (*p* < 0.05). The median pharmacokinetic estimates of the final multivariate model were CL/F = 1.2 L/h and V/F = 62.3 L.

The diagnostic plots and VPC confirmed the appropriateness of the model, as shown in Figure 1 and Figure 2, respectively. The individual-predicted concentrations and population-predicted concentrations based on the final model corresponded well with the observed concentrations. The typical value and standard error of the final model, CV, 95% confidence interval, and results of the bootstrap method were similar (Table 2). The VPC shows that most of the observed plots were between the 5th and 95th percentiles (Figure 2).

### 3.4. PTA Analysis

Monte Carlo simulations assuming a fixed dose of 200 mg or 400 mg/day were performed to indicate the breakpoint MIC in each scenario (Table 3). On the first dosing day (Day 1), the breakpoint MICs were 0.5–1 μg/mL for 200 mg/day and 1–2 μg/mL for 400 mg/day. On Days 8 and 15, breakpoint MICs were 1–2 μg/mL at 200 mg/day and 2–4 μg/mL at 400 mg/day. The breakpoint MICs changed by body weight and CLcr on Day 1. These changes were only caused by CLcr and not by bodyweight on Days 8 and 15.

To clarify the required doses satisfying *f* AUC/MIC = 50 for a target MIC = 2.0 μg/mL, additional Monte Carlo simulations were performed at 300, 400, 500, and 600 mg/day in each scenario. On Day 1, the required doses were proportional to body weight when CLcr was the same (Figure 3 and Appendix A). On Day 15, the required doses were proportional to CLcr when the body weight was the same (Figure 4 and Appendix A).

### 3.5. Verification of the Optimal Dose of Fluconazole

To determine the recommended fluconazole dose based on body weight and CLcr, Monte Carlo simulation was performed assuming a single loading dose on Day 1 and once daily maintenance doses after Day 2. The recommended doses by body weight and CLcr were 400–700 mg/day for the loading dose and 200–400 mg/day for the maintenance dose (Table 4). The maximum fluconazole concentration in blood did not exceed 80 μg/mL (data not shown).

## 4. Discussion

For the first time, we performed PPK analysis of fluconazole and built a PPK model in patients with hematological malignancy. Consequently, the Cockcroft-Gault equation–based CLcr was selected as a covariate for fluconazole clearance, and body weight was selected as the distribution volume. Based on the fact that 80% of administrated fluconazole is excreted from the kidneys as unchanged [30], it is reasonable that CLcr was selected as a covariate of CL/F. Previous reports have also shown that CLcr is a covariate of fluconazole clearance [18]. The distribution volume has been reported to be proportional to BMI in obese individuals [18]. In patients with liver cirrhosis, the distribution volume has been reported to decrease as expanded plasma and blood volume improve after liver transplantation [16]. The fluconazole distribution volume has also been reported to be identical to the whole-body water content [31], and it is reasonable to select body weight as a covariate of Vd/F. Because the protein binding rate of fluconazole was only 12%, albumin level was assumed to exert no effect on fluconazole concentration. Since gender was not considered as a covariate in previous reports [16,17,18,19,20], it was not used in this case also.

Vd/F and CL/F in this study were 62.3 L and 1.2 L/h, respectively. Several studies on the PPK analysis of fluconazole have been reported. According to the previous PPK analysis in healthy volunteers, Vd/F was reported as 55.4–59.4 L [32], which was slightly lower than our result. Cojutti et al. reported that Vd/F was 27.0 L in post-liver transplant patients and lower than other reports [16]. However, the reason for this finding has not been revealed. CL/F was reported as 0.55–0.79 L/h [16,19], which was lower than our result. Therefore, we showed for the first time that the CL/F of fluconazole is increased in hematological malignancies. The clearance of vancomycin and amikacin, which are excreted by the kidney, has been reported to be increased by ARC in patients with hematological malignancies [33,34]. Because vancomycin and amikacin are routinely recommended for therapeutic drug monitoring (TDM) [35,36], we usually monitor blood concentrations and modify their doses; In contrast, fluconazole is not recommended [37]. Therefore, monitoring of blood concentrations and modifying fluconazole doses are difficult in clinical settings. In addition, the prophylactic dose is often lower than the therapeutic dose [13,14]; however, a lower dose may not be sufficient for hematological malignancy whose CL/F increases.

We showed that fluconazole 400 mg/day achieved *f* AUC/MIC = 50 for up to MIC = 2 μg/mL at steady state; however, 200 mg/day was not achieved in the case of CLcr > 60 mL/min. Since the PK/pharmacodynamics (PD) parameter of fluconazole has been reported to be AUC/MIC [38], we adopted *f* AUC/MIC = 50, which Cojutti et al. [16] and Pai et al. [26] used, as our target PK/PD parameter. The target MIC was set at 2.0 μg/mL with reference to the sensitive norm of *Candida albicans* in CLSI M60 1st Edition [27] and previous epidemiological reports [28,29]. We simulated doses that could achieve PTA ≥ 90% for target PK/PD parameters (*f* AUC/MIC = 50 and MIC = 2 μg/mL). Fluconazole 400 mg/day prophylaxis was considered sufficiently feasible to reduce fungal infections in patients with hematological malignancies. Goodman et al. reported that fluconazole 400 mg/day was adequate to reduce fungal infections in bone marrow transplantation recipients [12]; our analysis would have validated their findings based on PK/PD theories. In contrast, fluconazole prophylaxis is often administered at 100–200 mg/day in Japan [13,14]; however, its prophylactic efficacy has not been validated. In PTA analysis, 200 mg/day administration failed to achieve target PK/PD parameters at steady state in those with CLcr > 60 mL/min. Therefore, it is suggested that administration of 200 mg/day may not provide adequate efficacy.

We determined the recommended dose based on body weight and CLcr and showed a loading dose of 400–700 mg/day and maintenance dose of 200–400 mg/day. The breakpoint MIC of fluconazole on the first day of administration was lower than the breakpoint MIC on the steady state. This may be because it takes time for blood concentrations to reach a steady state because the half-life of fluconazole is 31 h [39]. As the loading dose is recommended when fluconazole is used for treatment [7,40], a loading dose is recommended if adequate efficacy is needed from the first day. In addition, the safety of the recommended dose of fluconazole was investigated. Fluconazole is safe, with few adverse events occurring at 400 mg/day [3,11,12]. Previous reports have reported that convulsions occur when the blood fluconazole level is >80 μg/mL [41]. Simulated blood concentrations based on our nomogram did not exceed the maximum blood concentration of 80 μg/mL. The recommended dose in the steady state was never greater than 400 mg/day. Based on these results, our nomogram had a high safety profile. Our nomogram can provide adequate prophylaxis from the first day of administration, and it is possible to prevent low blood concentrations in patients with hematological malignancies, even without TDM.

This study has several limitations. First, this was a single-center study. It only included patients with hematological malignancies. We believe this is the first report to provide fluconazole pharmacokinetics information in patients with hematological malignancies whose ARC is a problem. Because the sample size and measurement points of this study were equal to or greater than previously published PPK analysis of fluconazole [16,17,18,19,20], this study has sufficient information that warrants the conduct of PPK analysis. Second, the optimal dosing regimen established in our study was not evaluated for efficacy and safety in clinical practice. Third, the target *f* AUC/MIC was 50 in this study. Although *f* AUC/MIC = 50 is widely used, there is little information regarding the *f* AUC/MIC of fluconazole. More research on the *f* AUC/MIC of fluconazole is needed. Fourth, the epidemiology of *Candida* infections varies across countries. A high target MIC is required to completely prevent breakthrough infection, which is unrealistic. Therefore, the target MIC was set with reference to the sensitive norm of *Candida albicans* in the CLSI M60 1st Edition. Even though there are several limitations, this is the first informative study to show the optimal dosing regimen based on Monte Carlo simulation for hematological malignancy patients. Robust evidence through prospective studies using these regimens must be obtained.

## 5. Conclusions

This PPK analysis included only patients with hematological malignancies, and the PPK model of hematological malignancy was built for the first time. Model-based Monte Carlo simulations suggested that fixed doses of fluconazole 200 mg/day may be ineffective with CLcr > 60 mL/min. Here, the optimal dosing regimens based on body weight and CLcr render fluconazole prophylaxis more reliable.

## Figures and Tables

**Figure 1 jof-07-00975-f001:**
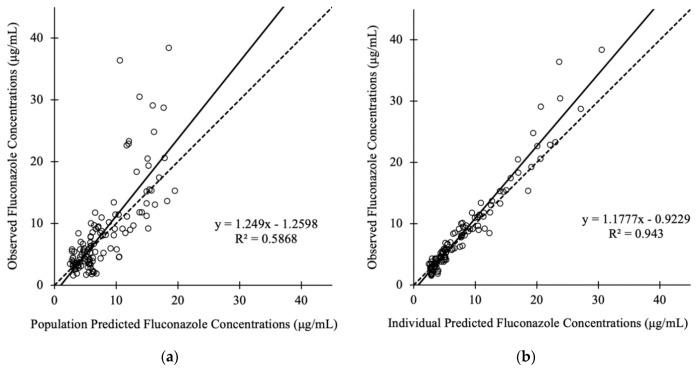
Diagnostic plot for the final covariate model. Figure 1 shows the (**a**) observed versus population-predicted concentrations and (**b**) observed versus individual-predicted concentrations in plasma. Solid lines refer to the linear regression between the observed and predicted concentrations. Dashed lines represent the identity lines between the observed and predicted concentrations.

**Figure 2 jof-07-00975-f002:**
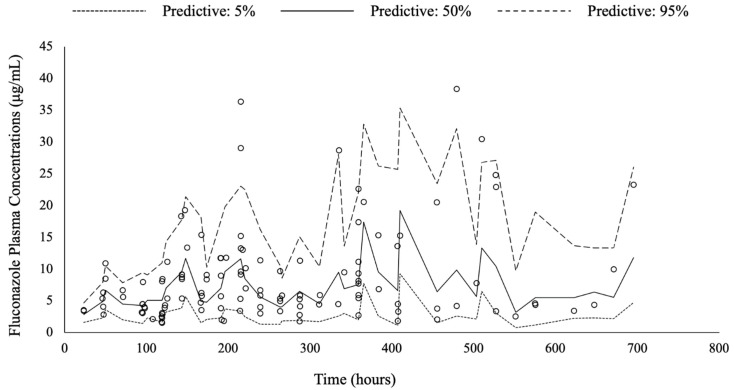
Prediction and variability corrected visual predictive check of final model. The black line at the center represents the median concentrations predicted, and the dotted black lines represent the 5th and 95th percentiles for the predictions. Each circle plot indicates the observed fluconazole plasma concentration.

**Figure 3 jof-07-00975-f003:**
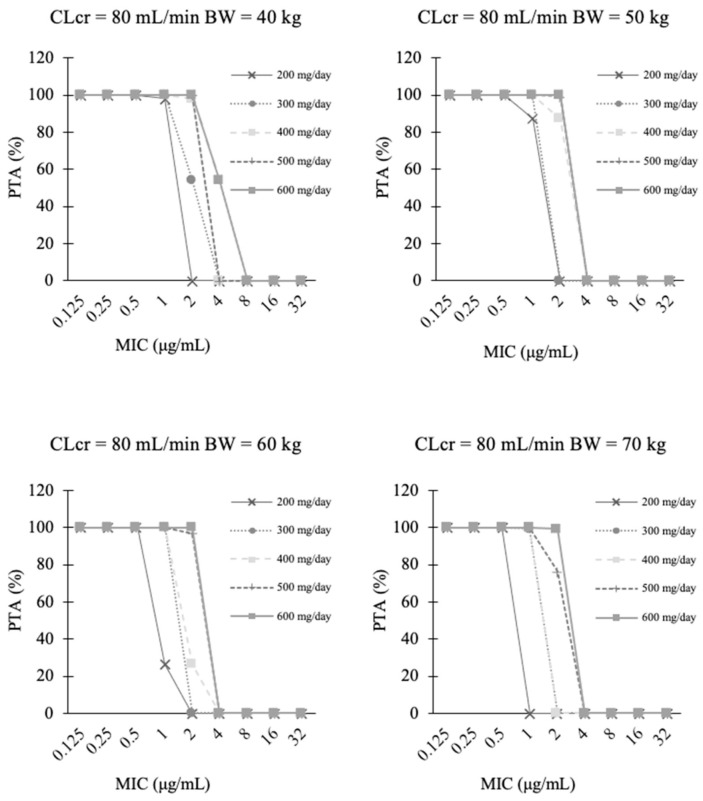
Probability of target attainment (PTA) of *f* AUC/MIC ratio ≥ 50 after fluconazole loading doses of 200 mg, 300 mg, 400 mg, 500 mg, and 600 mg administered on Day 1 (0–24 h). The target MIC was 2 μg/mL, and the CLcr (Cockcroft-Gault equation) was 80 mL/min. The other CLcr is shown in Appendix A.

**Figure 4 jof-07-00975-f004:**
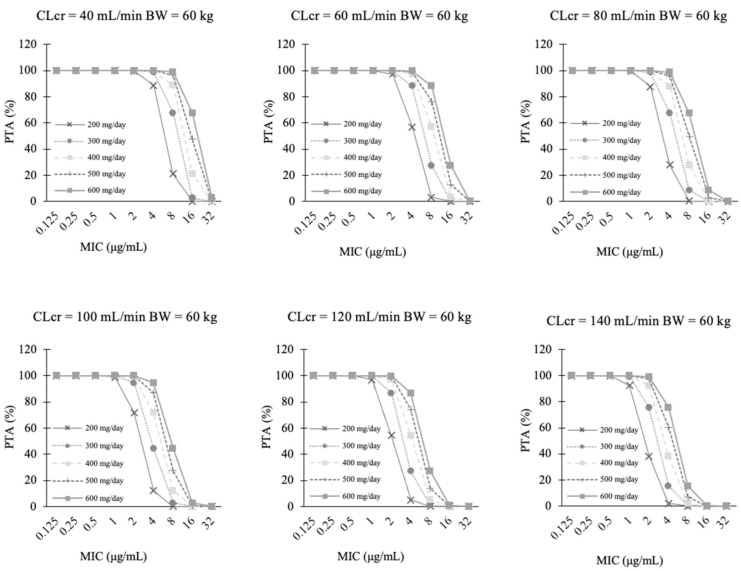
Probability of target attainment (PTA) of *f* AUC/MIC ratio ≥ 50 after fluconazole maintenance doses of 200 mg, 300 mg, 400 mg, 500 mg, and 600 mg administered on Day15 (336–360 h). Target MIC is 2 μg/mL, and Body Weight is 60 kg. Other Body Weight is shown in Appendix A.

**Table 1 jof-07-00975-t001:** Demographic and clinical characteristics of the population.

Characteristic	Median or Number	Range
Number of patients (male/female)	31/23	–
Body weight (kg)	57.6	39.8–99.1
Height (cm)	162.6	143.6–187.7
Age (years)	53	20–77
BMI (kg/m^2^)	21.7	15.8–29.1
BSA (m^2^)	1.62	1.31–2.23
CG-CLcr (mL/min)	87.1	31.1–193.6
eGFRcre (mL/min)	71.2	25.9–148.5
eGFRcre (mL/min/1.73 m^2^)	69.2	31.6–157.5
Disease		
Acute myeloid leukemia (AML)/Myelodysplastic syndromes (MDS)	10	–
Acute lymphoid leukemia (ALL)	6	–
non-Hodgkin lymphoma (NHL)	28	–
Hodgkin lymphoma (HL)	2	–
Multiple myeloma (MM)	3	–
Others	5	–
Treatment		
Chemotherapy	38	–
Autologous peripheral blood stem cell transplantation	9	–
Bone marrow transplantation	2	–
Cord blood transplantation	2	–
Allogeneic peripheral blood stem cell transplantation	1	–
Others	2	–

BSA, body surface area using the Dubois equation; CG-CLcr, estimated CLcr using the Cockcroft-Gault equation; eGFRcre, estimated GFR using the equation reported by Matsuo S et al.

**Table 2 jof-07-00975-t002:** Parameter estimates of fluconazole for the final covariate one-compartment model with first order input for oral administration.

Parameter	Final Model				Bootstrap Method *n* = 1000	
	Estimate	SE	CV (%)	95% CI	Median	95% CI
CL/F (L/h) = θ₁ × (CLcr/5.2)^θ^^₂^ × e^0.16^						
θ₁ (L/h)	1.03	0.08	7.38	0.88–1.18	1.03	0.86–1.19
θ₂ (L/h)	1.05	0.14	13.7	0.77–1.33	1.07	0.81–1.37
Vd/F (L) = θ₃ × (BW/57.6)^θ₄^						
θ₃ (L)	62.3	8.87	14.2	44.7–79.9	64.3	47.5–87.9
θ₄ (L)	1.06	0.36	34.4	0.34–1.78	1.08	0.08–2.24
ka (/h)	0.34	0.12	35.1	0.10–0.58	0.41	0.15–0.77

CLcr, estimated CLcr using the Cockcroft-Gault equation (L/h); BW, body weight (kg).

**Table 3 jof-07-00975-t003:** Fluconazole probability of target attainment (PTA) at different BW, CLcr, and loading dose regimen on Day 1, 8, and 15.

			PTA by MIC (μg/mL) and *f* AUC/MIC of 50																							
			Day 1 (0–24 h)								Day 8 (168–192 h)								Day 15 (336–360 h)							
Dose (mg/day)	BW (kg)	CLcr (mL/min)	0.125	0.25	0.5	1	2	4	8	16	0.125	0.25	0.5	1	2	4	8	16	0.125	0.25	0.5	1	2	4	8	16
200	40																									
		40	+	+	+	+	-	-	-	-	+	+	+	+	+	-	-	-	+	+	+	+	+	-	-	-
		60	+	+	+	+	-	-	-	-	+	+	+	+	+	-	-	-	+	+	+	+	+	-	-	-
		80	+	+	+	+	-	-	-	-	+	+	+	+	-	-	-	-	+	+	+	+	-	-	-	-
		100	+	+	+	+	-	-	-	-	+	+	+	+	-	-	-	-	+	+	+	+	-	-	-	-
		120	+	+	+	-	-	-	-	-	+	+	+	+	-	-	-	-	+	+	+	+	-	-	-	-
		140	+	+	+	-	-	-	-	-	+	+	+	+	-	-	-	-	+	+	+	+	-	-	-	-
	50																									
		40	+	+	+	+	-	-	-	-	+	+	+	+	+	-	-	-	+	+	+	+	+	-	-	-
		60	+	+	+	+	-	-	-	-	+	+	+	+	+	-	-	-	+	+	+	+	+	-	-	-
		80	+	+	+	-	-	-	-	-	+	+	+	+	-	-	-	-	+	+	+	+	-	-	-	-
		100	+	+	+	-	-	-	-	-	+	+	+	+	-	-	-	-	+	+	+	+	-	-	-	-
		120	+	+	+	-	-	-	-	-	+	+	+	+	-	-	-	-	+	+	+	+	-	-	-	-
		140	+	+	+	-	-	-	-	-	+	+	+	+	-	-	-	-	+	+	+	+	-	-	-	-
	60																									
		40	+	+	+	-	-	-	-	-	+	+	+	+	+	-	-	-	+	+	+	+	+	-	-	-
		60	+	+	+	-	-	-	-	-	+	+	+	+	+	-	-	-	+	+	+	+	+	-	-	-
		80	+	+	+	-	-	-	-	-	+	+	+	+	-	-	-	-	+	+	+	+	-	-	-	-
		100	+	+	+	-	-	-	-	-	+	+	+	+	-	-	-	-	+	+	+	+	-	-	-	-
		120	+	+	+	-	-	-	-	-	+	+	+	+	-	-	-	-	+	+	+	+	-	-	-	-
		140	+	+	+	-	-	-	-	-	+	+	+	+	-	-	-	-	+	+	+	+	-	-	-	-
	70																									
		40	+	+	+	-	-	-	-	-	+	+	+	+	+	-	-	-	+	+	+	+	+	-	-	-
		60	+	+	+	-	-	-	-	-	+	+	+	+	+	-	-	-	+	+	+	+	+	-	-	-
		80	+	+	+	-	-	-	-	-	+	+	+	+	-	-	-	-	+	+	+	+	-	-	-	-
		100	+	+	+	-	-	-	-	-	+	+	+	+	-	-	-	-	+	+	+	+	-	-	-	-
		120	+	+	+	-	-	-	-	-	+	+	+	+	-	-	-	-	+	+	+	+	-	-	-	-
		140	+	+	+	-	-	-	-	-	+	+	+	+	-	-	-	-	+	+	+	+	-	-	-	-
400	40																									
		40	+	+	+	+	+	-	-	-	+	+	+	+	+	+	-	-	+	+	+	+	+	+	-	-
		60	+	+	+	+	+	-	-	-	+	+	+	+	+	+	-	-	+	+	+	+	+	+	-	-
		80	+	+	+	+	+	-	-	-	+	+	+	+	+	-	-	-	+	+	+	+	+	-	-	-
		100	+	+	+	+	+	-	-	-	+	+	+	+	+	-	-	-	+	+	+	+	+	-	-	-
		120	+	+	+	+	-	-	-	-	+	+	+	+	+	-	-	-	+	+	+	+	+	-	-	-
		140	+	+	+	+	-	-	-	-	+	+	+	+	+	-	-	-	+	+	+	+	+	-	-	-
	50																									
		40	+	+	+	+	+	-	-	-	+	+	+	+	+	+	-	-	+	+	+	+	+	+	-	-
		60	+	+	+	+	+	-	-	-	+	+	+	+	+	+	-	-	+	+	+	+	+	+	-	-
		80	+	+	+	+	-	-	-	-	+	+	+	+	+	-	-	-	+	+	+	+	+	-	-	-
		100	+	+	+	+	-	-	-	-	+	+	+	+	+	-	-	-	+	+	+	+	+	-	-	-
		120	+	+	+	+	-	-	-	-	+	+	+	+	+	-	-	-	+	+	+	+	+	-	-	-
		140	+	+	+	+	-	-	-	-	+	+	+	+	+	-	-	-	+	+	+	+	+	-	-	-
	60																									
		40	+	+	+	+	-	-	-	-	+	+	+	+	+	+	-	-	+	+	+	+	+	+	-	-
		60	+	+	+	+	-	-	-	-	+	+	+	+	+	+	-	-	+	+	+	+	+	+	-	-
		80	+	+	+	+	-	-	-	-	+	+	+	+	+	-	-	-	+	+	+	+	+	-	-	-
		100	+	+	+	+	-	-	-	-	+	+	+	+	+	-	-	-	+	+	+	+	+	-	-	-
		120	+	+	+	+	-	-	-	-	+	+	+	+	+	-	-	-	+	+	+	+	+	-	-	-
		140	+	+	+	+	-	-	-	-	+	+	+	+	+	-	-	-	+	+	+	+	+	-	-	-
	70																									
		40	+	+	+	+	-	-	-	-	+	+	+	+	+	+	-	-	+	+	+	+	+	+	-	-
		60	+	+	+	+	-	-	-	-	+	+	+	+	+	+	-	-	+	+	+	+	+	+	-	-
		80	+	+	+	+	-	-	-	-	+	+	+	+	+	-	-	-	+	+	+	+	+	-	-	-
		100	+	+	+	+	-	-	-	-	+	+	+	+	+	-	-	-	+	+	+	+	+	-	-	-
		120	+	+	+	+	-	-	-	-	+	+	+	+	+	-	-	-	+	+	+	+	+	-	-	-
		140	+	+	+	+	-	-	-	-	+	+	+	+	+	-	-	-	+	+	+	+	+	-	-	-

CLcr, estimated CLcr using the Cockcroft-Gault equation; BW, body weight (kg). A plus sign indicates that at least 90% of fluconazole PTA is achieved; a minus sign (shaded) indicates that fluconazole PTA attainment failed to achieve 90%.

**Table 4 jof-07-00975-t004:** Nomogram of fluconazole dose for prophylaxis.

		CLcr (Cockcroft-Gault Equation)				
		40–60 mL/min	60–80 mL/min	80–100 mL/min	100–120 mL/min	120–140 mL/min
Loading dose (Required in the situation neutropenia is predicted within a week)						
Body	40–50 kg	400 mg/day	500 mg/day	500 mg/day	500 mg/day	600 mg/day
Weight	50–60 kg	500 mg/day	500 mg/day	500 mg/day	600 mg/day	600 mg/day
	60–70 kg	500 mg/day	600 mg/day	600 mg/day	600 mg/day	700 mg/day
Maintenance dose						
Body	40–50 kg	200 mg/day	300 mg/day	300 mg/day	400 mg/day	400 mg/day
Weight	50–60 kg	200 mg/day	300 mg/day	300 mg/day	400 mg/day	400 mg/day
	60–70 kg	200 mg/day	300 mg/day	300 mg/day	400 mg/day	400 mg/day

## Data Availability

The data presented in this study are available on request from the corresponding author. The data are not publicly available due to privacy.

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
