# Peer review of "Population Pharmacokinetic Analysis and Dosing Optimization of Prophylactic Fluconazole in Japanese Patients with Hematological Malignancy"

_jof, 2021, doi:10.3390/jof7110975_

Round 1
Reviewer 1 Report
The manuscript entitled “Population Pharmacokinetic Analysis and Dosing Optimization of Fluconazole Prophylaxis in Hematological Malignancy Patients” by Sakamoto Y., et al presented data related to fluconazole PPK in Japanese patients. The manuscript was well-written and it is easy to follow. There are some points need to clarify or revise.
- Title - This is a study with a small sample size in Japanese patients and the data may not be applied to use in other center where the epidemiology of fungal infections may be different. Therefore, the title should be clarified that this study was conducted in a single center in Japan.
- Introduction – The authors try to emphasize that deep-seated mycosis is important in hematological patients. However, the common invasive fungal diseases in hematological patients is mold infection (such as invasive aspergillosis), and fluconazole does not have activity against molds. Fluconazole prophylaxis can be effective for yeast (such as Candida) infection only. The authors should revise the introduction section, as well as the reference, to highlight Candida infection in hematological patients rather than mentioning deep-seated mycosis as a whole.
- Fluconazole can bind to protein and has drug-drug interaction with immunosuppressive agent. Should the authors include albumin, immunosuppressive agents/chemotherapy and gender as covariates in the simulation? If not, what is the reason? Please mention in the discussion.
- The epidemiology of Candida infection in Japan is different from other countries. For example, India and Southeast Asia are highly prevalent with Candida tropicalis which is resistant to fluconazole (MIC > 32). Also, Candida glabrata is prevalent in the US with MIC to fluconazole > 64. Therefore, the recommendation derived from this study would be applicable to Japan only and may not be able to apply to other part of the world which exhibits different epidemiology. This should be discussed in the manuscript (may be as a limitation).
Author Response
Reviewer Comments:
Reviewer 1: The manuscript entitled “Population Pharmacokinetic Analysis and Dosing Optimization of Fluconazole Prophylaxis in Hematological Malignancy Patients” by Sakamoto Y., et al presented data related to fluconazole PPK in Japanese patients. The manuscript was well-written and it is easy to follow. There are some points need to clarify or revise.
- Title - This is a study with a small sample size in Japanese patients and the data may not be applied to use in other center where the epidemiology of fungal infections may be different. Therefore, the title should be clarified that this study was conducted in a single center in Japan.
Response:
Thank you for your comment. This study was conducted only on Japanese patients. As mentioned in the limitations section, this was a single-center study. However, since the target MIC was set at 2.0 μg/mL with reference to the sensitive norm of Candida albicans in CLSI M60 1st Edition, we believe that the epidemiology of fungal infections does not affect our conclusion. In addition, patient characteristics such as body weight, height, BSA, and BMI were varied. Therefore, we revised our title as follows:
‘Population Pharmacokinetic Analysis and Dosing Optimization of Prophylactic Fluconazole in Japanese Patients with Hematological Malignancy’
If the reviewer considers that the Title should contain the term ‘single center, we will be glad to revise it accordingly.
- Introduction – The authors try to emphasize that deep-seated mycosis is important in hematological patients. However, the common invasive fungal diseases in hematological patients is mold infection (such as invasive aspergillosis), and fluconazole does not have activity against molds. Fluconazole prophylaxis can be effective for yeast (such as Candida) infection only. The authors should revise the introduction section, as well as the reference, to highlight Candida infection in hematological patients rather than mentioning deep-seated mycosis as a whole.
Response:
We appreciate your suggestions. We have revised our main text as follows.
Invasive candidiasis is one of the most common fungal infections and includes candidemia and deep-seated candidiasis[1]. Hematological malignancy is a risk factor for invasive candidiasis, and prophylaxis of this is a critical priority [2–4]. In a report of autopsy cases in Japan, deep-seated candidiasis was found to be the direct cause of death in 66.7% of patients with leukemia or myelodysplastic syndrome and concomitant deep-seated candidiasis [5]. According to a report on post-hematopoietic stem cell transplant recipients in the United States, one-year survival rate was only 33.6% in these patients [6]. Therefore, prevention of invasive candidiasis is a key challenge. (lines 45–49)
We have revised ‘deep-seated mycosis’ to ‘invasive candidiasis’ for better readability. (lines 19–20, 36, 69–70)
- Fluconazole can bind to protein and has drug-drug interaction with immunosuppressive agent. Should the authors include albumin, immunosuppressive agents/chemotherapy and gender as covariates in the simulation? If not, what is the reason? Please mention in the discussion.
Response:
We appreciate your comment. The protein binding rate of fluconazole was 12%; therefore, we did not evaluate albumin level as a covariate of fluconazole pharmacokinetic parameters. In general, fluconazole inhibits CYP3A4 and affects the pharmacokinetics of immunosuppressive agents and chemotherapy. In contrast, the pharmacokinetics of fluconazole were not affected by these agents; thus, we have not evaluated these interactions. Gender was not a covariate in previous reports; therefore, we did not use gender as a covariate for the evaluation of fluconazole pharmacokinetic parameters. As per your suggestion, we have revised our main text as follows:
‘Because the protein binding rate of fluconazole was only 12%, albumin level was assumed to exert no effect on fluconazole concentration. Since gender was not considered as a covariate in previous reports, it was not used in this case also [16-20].’ (lines 250–261)
- The epidemiology of Candida infection in Japan is different from other countries. For example, India and Southeast Asia are highly prevalent with Candida tropicalis which is resistant to fluconazole (MIC > 32). Also, Candida glabrata is prevalent in the US with MIC to fluconazole > 64. Therefore, the recommendation derived from this study would be applicable to Japan only and may not be able to apply to other part of the world which exhibits different epidemiology. This should be discussed in the manuscript (may be as a limitation).
Response:
Thank you for your valuable comment. Indeed, it would be best if fluconazole prophylaxis could completely prevent breakthrough infections. However, to achieve complete prevention of Candida infection, including those caused by fluconazole low-susceptible Candida species, high blood concentrations of fluconazole are required, resulting in a risk of side effects. Therefore, the target MIC was set at 2 µg/mL according to CLSI M60 to cover Candida species that are sensitive to fluconazole. However, since these are considered limitations, we have revised our main text as follows:
‘Fourth, the epidemiology of Candida infections varies across countries. A high target MIC is required to completely prevent breakthrough infection, which is unrealistic. Therefore, the target MIC was set with reference to the sensitive norm of Candida albicans in CLSI M60 1st Edition.’ (lines 326–329)
Reviewer 2 Report
An extremely interesting and useful idea to establish the optimal prophylactic dose of fluconazole in haematological malignancy patients. It is also important to check whether the 200 mg dose used in prophylaxis is effective.
I think however that the material and method part is not detailed enough. Fluconazole was administered 200 mg/day. Was it a single dose or was it administered multiple days? Were blood samples taken only on the first day of administration? In the methodology part the authors state that Blood samples were collected at 1 hour before (trough levels) and at 2, 4, and 12 hours after fluconazole administration, at available times. The total number of patients is 54. However, only 125 samples were collected. How was this number of samples arrived at?
I think the authors should revise the English wording to make the article more readable.
Author Response
Reviewer Comments:
Reviewer 2: An extremely interesting and useful idea to establish the optimal prophylactic dose of fluconazole in haematological malignancy patients. It is also important to check whether the 200 mg dose used in prophylaxis is effective.
- I think however that the material and method part is not detailed Fluconazole was administered 200 mg/day. Was it a single dose or was it administered multiple days? Were blood samples taken only on the first day of administration? In the methodology part the authors state that Blood samples were collected at 1 hour before (trough levels) and at 2, 4, and 12 hours after fluconazole administration, at available times. The total number of patients is 54. However, only 125 samples were collected. How was this number of samples arrived at?
Response:
Thank you for your valuable comment. Blood samples were not collected at all times, and it depended on the physical stress on the patients and the feasibility of clinical staff. As per your suggestion, we have revised our main text as follows:
‘Fluconazole was administered orally at a dose of 200 mg once daily every 24 h. Blood samples were collected at 1−4 points at indicated time (1 h before (trough levels), 2, 4, or 12 h) after fluconazole administration.’ (lines 98–100)
- I think the authors should revise the English wording to make the article more readable.
Response:
The manuscript has re-checked and edited by a native English speaker (Editage).
